# Daily Variation in the Feeding Activity of Pacific Crown-of-Thorns Starfish (*Acanthaster* cf. *solaris*)

**DOI:** 10.3390/biology14081001

**Published:** 2025-08-05

**Authors:** Josie F. Chandler, Deborah Burn, Will F. Figueira, Peter C. Doll, Abby Johandes, Agustina Piccaluga, Morgan S. Pratchett

**Affiliations:** 1College of Science and Engineering, James Cook University, Townsville, QLD 4811, Australia; deborah.burn1@my.jcu.edu.au (D.B.); peter.doll@jcu.edu.au (P.C.D.); abby.johandes@my.jcu.edu.au (A.J.); morgan.pratchett@jcu.edu.au (M.S.P.); 2Australian Institute of Marine Science, Townsville, QLD 4810, Australia; 3School of Life and Environmental Sciences, University of Sydney, Sydney, NSW 2006, Australia; will.figueira@sydney.edu.au (W.F.F.); agustina.piccaluga@sydney.edu.au (A.P.)

**Keywords:** behaviour, coral reefs, disturbance, management

## Abstract

Crown-of-thorns starfish are coral-eating predators that can cause widespread damage to coral reefs when their populations grow out of control. To understand their impacts and improve management of population outbreaks, it is necessary to understand how much coral each starfish eats and how often they feed. In this study, we closely tracked the daily feeding activity of eight individual starfish on the Great Barrier Reef using advanced 3D modelling techniques. We found that CoTS do not feed every day, often going 2–3 days without eating, and that feeding amounts varied greatly between individuals. On average, each starfish consumed about one coral colony per day, mostly from the *Acropora* group of corals, which are both the preferred food and the most structurally complex. We also measured how long the feeding scars stayed visible before becoming overgrown with algae, which is important for interpreting their feeding behaviour during rapid large-scale reef surveys. These results improve our understanding of CoTS feeding behaviour and allow us to better estimate and model their impacts on coral reefs, ultimately aiding reef conservation and management efforts.

## 1. Introduction

Climate change, particularly the occurrence of severe marine heatwaves and corresponding mass coral bleaching, poses the foremost threat to coral reefs [1]. However, population outbreaks of the coral predator crown-of-thorns starfish (CoTS, *Acanthaster* spp.) remain a major contributor to coral loss throughout the Indo-Pacific [2,3]. *Acanthaster* cf. *solaris* is a species of CoTS, natively distributed throughout the tropical Indo Pacific, including the Great Barrier Reef. Since the 1960s there have been four major CoTS outbreaks on the GBR [4,5,6,7] with renewed outbreaks of CoTS detected in the northern GBR since 2021 [8,9]. Management of CoTS outbreaks represents the foremost opportunity to redress widespread and sustained coral loss [10,11,12], and thereby maximise opportunities for natural adaptation among coral populations and communities to changing environmental conditions.

The ecological impacts of CoTS are fundamentally driven by their feeding behaviour, not only in terms of the overall extent of coral loss that occurs during major outbreaks (e.g., [13]), but also shifts in the structure of coral assemblages due to highly selective feeding patterns [14]. In general, CoTS have strong feeding preferences for corals from the Acropororidae and Pocilloporidae families, whilst Poritidae and Merulinidae corals are non-preferred [15,16,17,18,19,20]. The evolution of these strong prey preferences is thought to be determined by a multitude of factors, including accessibility and availability of different coral prey, as well as the nutritional benefit to CoTS [15].

Despite extensive research into prey preferences, empirical data on individual feeding rates of CoTS is limited. Most existing estimates of CoTS feeding rates come from studies conducted between the 1960s and 1990s, which provide only coarse approximations from single linear measurements of feeding scars and prey corals (e.g., maximum diameter), rather than accounting for the full extent of coral consumption. Reported feeding rates vary widely, with estimates ranging from 116–187 cm^2^ day^−1^ [20] to 260–734 cm^2^ day^−1^ [21] with additional values of 66–478 cm^2^ day^−1^ [22], 150 cm^2^ day^−1^ [23], 160–320 cm^2^ day^−1^ [24] and 350 cm^2^ day^−1^ [25]. In many cases, feeding rates were derived from linear measurements converted to two-dimensional (2D) projected areas, which do not capture the structural complexity of corals. Recent studies (e.g., [21]) have used software such as ImageJ to estimate projected planar areas with greater resolution; however, these methods still do not account for the three-dimensional (3D) structure of corals, including height and complexity, which may influence feeding estimates. Only Caballes et al. [26] have taken measurements of the 3D areal extent of coral consumed by CoTS, which was conducted during a laboratory experiment.

Feeding rates of CoTS are generally expressed as the average area of coral that is consumed per individual starfish per day (e.g., [21,22]) with limited consideration of variation in feeding activity and rates over successive days, or between individual starfish. Whilst these average feeding rates are useful for assessing potential coral loss with increasing densities of CoTS, they overlook potentially important information on individual differences in feeding behaviour. For example, it is not known whether CoTS exhibit characteristic grazing habits, whereby the amount of coral consumed is relatively consistent from day to day and limited by the local abundance and distribution of prey corals. Conversely, CoTS may exhibit a ‘feast and famine’ feeding strategy (*sensu* [27]), whereby periods of intensive feeding activity may be followed by periods of inactivity. Establishing the predominant feeding habits can provide important insights into whether CoTS are food-limited and how they regulate feeding activity to meet energetic demands. This distinction is critical for predicting how CoTS populations may respond under future conditions when coral prey may become increasingly scarce [28,29], and metabolic demands may rise with ocean warming [30].

One of the foremost factors known to influence feeding behaviour, including rate of consumption, is the availability of different coral prey [18,21]. Higher feeding rates have been recorded for *Acropora* corals in the laboratory [26] and in habitats that are *Acropora*-dominated [21], with suppressed feeding rates recorded on *Porites* [26]. These differential feeding rates strongly reflect known differences in prey preferences. However, these prey preferences cannot be fully explained by differences in coral tissue composition or nutritional value (e.g., [22,26]) and may instead reflect variation in overall feeding efficiency [15,31]. More specifically, the amount of nutrients extracted during individual feeding bouts, whereby CoTS evert their stomach and simultaneously digest coral tissues in direct contact with their stomach folds [32], would vary with size and complexity of different corals [33], and thereby the overall tissue surface area. Incorporating 3D surface area into feeding estimates could therefore yield more accurate assessments of energetic intake and better inform models of population dynamics and coral reef impact. While the logistical challenges of quantifying 3D structure in situ have historically limited such efforts, recent advances in underwater close-range photogrammetry [34,35] have afforded new opportunities to obtain detailed measurements of tissue surface area, even for morphologically complex corals.

In addition to characterising rates and variability in feeding behaviour, understanding the temporal persistence of feeding scars is essential for interpreting signs of recent feeding in ecological surveys. The time it takes for feeding scars to transition from freshly consumed to being overgrown with turf algae, referred to as ‘scar ageing’, is rarely quantified. However, this process is important for estimating feeding activity in large-scale assessments such as the Scooter-assisted large-area diver-based (SALAD) surveys [8], where visible scars are used to infer recent feeding activity. Few studies have empirically determined the timeframe over which scars remain detectable, limiting the accuracy of such estimates. Together, high-resolution measurements of daily feeding rates and scar ageing dynamics can improve predictions of CoTS impacts and enhance the resolution of ecological monitoring tools.

This study used close-range photogrammetry to provide highly resolved estimates of daily feeding rates of individual CoTS, whilst subsequently monitoring progression of scar ageing on recently consumed corals. This data will help refine existing estimates of daily consumption and, by incorporating a three-dimensional perspective, offer new insights into the reliance of CoTS on different coral genera and their energetic intake under mixed-species diets. Daily tracking of individual CoTS also provides unprecedented insights into the temporal variability of feeding activity, enhancing our understanding of feeding strategies and potential ecological constraints. Additionally, tracking the rate at which feeding scars are obscured by algal overgrowth ensures more accurate interpretation of recent feeding activity in monitoring programmes.

## 2. Materials and Methods

This study was conducted at Lizard Island (14°40′ S, 145°27′ E) in the northern GBR, Australia (Figure 1a,b), during the austral summer (November–December) of 2023. Lizard Island was selected as the study site due to its well-documented CoTS populations and the availability of well-equipped field facilities (Lizard Island Research Station), which enabled the repeated daily observations required for this study.

All observations were conducted on reef slope habitats at Lizard Island. Survey sites (Granite Bluff, Turtle Beach and Casuarina) were selected to ensure broadly consistent environmental conditions, including similar depths (4–10 m), low to moderate wave exposure, and high coral cover (30–50%) with similar coral assemblages and reef structure. Eight adult *Acanthaster* cf. *solaris* starfish (>30 cm diameter), which are native to this region, were selected for tracking during this study, each separated by at least 50 m. While no physical tags were attached to the starfish, individuals were reliably identified based on a combination of body size, distinct morphological features and their spatial position within the reef, which was recorded daily. Given the limited daily displacement of *Acanthaster* cf. *solaris*, especially in high coral cover environments, consistent relocation and identification of individuals across consecutive survey days was readily achievable. To minimise the likelihood of overlapping home ranges, a 50 m radius around each starfish was searched for conspecifics prior to commencement of the study. Given that CoTS are known to exhibit generally small home ranges when food resources are readily available [18], it was anticipated that CoTS at these high coral cover sites would not overlap during the study.

The focal CoTS were tracked for between 9 and 13 days during a 13-day survey period to quantify live coral consumption over successive 24-h periods. Two individuals were only tracked for 9 days due to logistical constraints. Additionally, no observations were conducted on two survey days (25 November and 30 November) due to diving limitations. For these instances (24–26 November and 29 November–1 December), coral consumption recorded over the 48 h was halved to estimate daily feeding rates (see Figure 2). For analyses comparing cumulative feeding among individuals, only the 9 days for which all individuals had complete data were used to ensure consistency across all CoTS (Figure 2).

On Day 0, a thorough search of the localised reef area was conducted and all feeding scars that were present in the vicinity of each of the focal starfish were marked with blank cattle tags. The location of the starfish was also marked with a tag (‘Day 0’) and recorded on a hand-drawn site map. Each site was revisited every 24–48 h and any new scars that were present were counted, identified to genus, measured (maximum diameter), 3D modelled and marked with numbered tags. The specific location of the starfish was also marked with ‘Day X’ tags as well as on the site map. Focal CoTS were easily identified based on the known size of individuals and proximity to the previous day’s position.

Daily consumption of coral was calculated for each starfish based on the average number of distinct colonies and the cumulative areal extent of fresh feeding scars for each 24-h period. The total number of coral colonies consumed per day was recorded in situ while photogrammetry was used to estimate both the planar (2D) area and total (3D) surface area of each distinct feeding scar (see below for feeding scar 3D modelling methods).

All CoTS tracked in this study were measured to the nearest centimetre to record maximum body diameter. At the conclusion of the study, individuals were dissected and weighed to determine (a) sex and (b) Gonadosomatic Index (GSI), used as a proxy for reproductive investment. GSI was calculated as the gonad weight divided by the total body weight, multiplied by 100. These data were captured to identify whether these intrinsic factors may be significantly influencing individual feeding rates.

### 2.1. Diet Composition

The prey coral for each feeding scar in the study was identified to genus in situ. However, for comparison of CoTS diets, corals were grouped by morphotaxa. Nine common genera were selected based on their relative abundance in the environment and some genera, namely *Acropora*, *Montipora* and *Porites*, were refined further into morphotaxa groups. The resulting 11 groups were: *Acropora*—table, *Acropora*—staghorn, *Acropora*—other, *Astreopora*, *Galaxea*, *Goniastrea*, *Lobophyllia*, *Montipora*—encrusting, *Porites*—massive, *Pocillopora* and *Stylophora*. Three further groups: Other—branching, Other—massive and Other—encrusting were included to encompass all other possible corals, where branching was defined as colonies with numerous branches, usually including secondary branches which are smaller offshoots growing from the primary branches, massive was defined as colonies with hemispherical growth and encrusting was defined as colonies with low relief that followed the contour of the reef.

### 2.2. Scar Ageing

To capture the ageing process of CoTS feeding scars, fresh feeding scars known to be <24 h old were photographed alongside a CoralWatch card [36] and then photographed every day in the same way for the duration of the study. Changes in colour intensity of corals, as algae progressively colonised coral skeletons, were used to quantify the process of scar ageing. Photographs of feeding scars were taken from a consistent distance of 30 cm using an Olympus TG-6 camera (Olympus, Sydney, Australia) without flash, with the white balance adjusted separately for each scar, using a white slate for reference.

The time taken for a scar to progress beyond the colour of category B2 on the CoralWatch card was quantified, as this threshold has previously been used as the point of exclusion for recording feeding scars in SALAD surveys [8]. To objectively determine when a scar surpassed this threshold, the CIELAB colour space (L*a*b*) was used to measure changes in lightness and chroma (colour intensity). This quantitative approach helps to minimise observer bias.

L*a*b* values were extracted using the colour dropper tool in Adobe Lightroom (Version 8.4), where L* indicates lightness, a* represents the red–green axis and b* the yellow–blue axis. These values were then compared to known L*a*b* values for category B2 on the CoralWatch card (a* = −6, b* = 16). Given that pure white in CIELAB space has values of a* = 0 and b* = 0, any scar colour with a* < −7 or b* > 16 was considered to have exceeded the B2 threshold, indicating significant ageing.

This method was applied to nine time series of scar photographs, representing three different coral morphologies across three distinct sites. For each scar an “ageing factor” was calculated, defined as the number of days required for the scar’s colour to surpass the B2 threshold. These values were then averaged across all nine scars to determine the mean ageing factor.

### 2.3. Three-Dimensional Modelling of Feeding Scars

Gopro Hero 11 Black cameras (GoPro, Sydney, Australia) were used to collect close-range imagery of individual colonies for 3D modelling. Structure from Motion techniques (SfM) described by Ferrari et al. [34] were followed; however, 4K video footage (30 fps) was used instead of timelapse imagery due to faster in-water data collection capability, allowing for more replicates per morphotaxon [37]. Additionally, custom-built scaling objects (Agisoft Metashape coded targets, 12 bit, inverted, 20% of full size, target centres 60 mm apart) were incorporated into the colony scene. Image acquisition involved filming the colony from all angles, moving in arcs from top to bottom. Videos were filmed in natural lighting. Jpeg images were extracted from every fourth frame of the 4k videos. Image sets ranged from 150 to 1400 images, depending on the length of the video (averaged ~1 min) which was related to coral size and complexity.

Three-dimensional models were reconstructed from image sets using Agisoft Metashape (v 1.7.6) software. Any extraneous mesh material that was not part of the colony was removed using Agisoft Metashape selection tools. Models were scaled using at least two sets of independent scale bars (60 mm in length) from the coded target objects described above. Resulting models had an average mesh resolution (distance between vertices) of 1.86 mm and an average ground sample distance of 0.18 mm/pixel. Models were cropped as (a) the entire coral colony and (b) the scar only. The colony 3D area was extracted using the Mesh-Area tool in Agisoft Metashape. In order to extract a 2D area, the model must first be projected onto a 2D plane in a manner similar to taking a top-down photograph of the coral. Thus, the models were visually orientated in the software to replicate a top-down field of view, relative to the scar surface, where the z-axis was aligned perpendicular to this view plane. Once this was completed, the 2D area of the colony and the scars were estimated using the Export Report tool within the software with projection set to the X-Y plane.

### 2.4. Statistical Analysis

To investigate the factors influencing daily feeding rates, we modelled daily feeding rates (planar area consumed in cm^2^) using a zero-inflated generalised linear mixed-effects model (ZIGLMM). A Gamma distribution was used to model the continuous, positive nature of daily feeding rates, while a zero-inflated component was included to handle the high proportion (32.3%) of days with no observed feeding. An assumption of this model is that zeros arise from true non-events (i.e., genuine non-feeding days), rather than from measurement error or detection failure.

The response variable was the daily planar feeding area (cm^2^). Fixed effects were starfish body size (diameter in cm) and gonadosomatic index (GSI), both of which may influence energetic demand or feeding motivation. Random effects were included for individual identity (Cot_no), to account for repeated measures and inter-individual variation in feeding behaviour. Site was also included as a random effect to control for differences in environmental conditions, such as coral availability, across locations.

The model was fitted using the glmmTMB package (1.1.7) [38] in R with a log link function for the conditional (non-zero) component, and a logit link for the zero-inflation component. Model diagnostics, including residual simulations using the DHARMa package (0.4.6) [39], indicated no major violations of distributional assumptions or overdispersion, and confirmed an adequate model fit. Ref. [39] Box plots and stacked bar plots were computed using the ggplot2 package (3.5.1) in R [40].

## 3. Results

### 3.1. Feeding Patterns

CoTS did not feed on 35% of the nights, averaged across all focal individuals (Figure 2). On occasions when no feeding was observed, CoTS were located in the same location during successive daily observations, suggesting that no movement took place; but no CoTS remained inactive for more than 3 consecutive days. Days of inactivity were often consecutive, with periods usually lasting 2–3 days.

### 3.2. Feeding Rates

The mean daily feeding rates for individual CoTS was 1.35 coral colonies (±0.18 SE) per day or 198.4 cm^2^ day^−1^ (±23.86 SE) planar area per day or 998.83 cm^2^ day^−1^ (±128.69 SE) total 3D surface area per day, accounting for the multiple days during the survey period where no feeding was recorded.

Daily consumption values were highly variable both within and among starfish (Figure 3), ranging from 0 to 9 colonies a day, 0 cm^2^ day^−1^ to 1261.0 cm^2^ day^−1^ (planar area) and 0 cm^2^ day^−1^ to 5242.8 cm^2^ day^−1^ (3D surface area). However, when averaged over the whole survey period, individual mean feeding rates were relatively consistent for most CoTS, especially for planar area of coral consumed with 5 out of 8 CoTS having individual mean feeding rates between 115–200 cm^2^ day^−1^ (Figure 3).

To investigate potential drivers of this variation, we applied a zero-inflated generalised linear mixed-effects model (ZIGLMM), which indicated that individual identity (CoTS ID) accounted for a substantial proportion of the observed variation in feeding behaviour (random intercept SD = 0.25), while site-level differences were negligible (SD < 0.001). Neither body size (*p* = 0.146) nor GSI (*p* = 0.349) significantly influenced variation in feeding rate on feeding days. This suggests that variation in feeding behaviour was not strongly driven by the intrinsic factors measured here, but instead reflects individual-level behavioural variability. Sex was not included as a predictor due to the strongly skewed sex ratio (7 of 8 individuals were female).

When daily surface area consumption was summed over the 9-day survey period for each starfish (effectively representing cumulative tissue consumption), total surface area consumption was highly variable among individuals, ranging from 3897.4 cm^2^ (CoTS 4) to 20,036.9 cm^2^ (CoTS 3; Figure 4b).

### 3.3. Diet Composition

A total of 17 different coral genera were consumed during this study, which were assigned to 15 morphotaxa groups. *Acropora*—other colonies were consumed in excess of all other colonies for six out of eight CoTS (Figure 4a), with a maximum of 12 *Acropora-* other colonies consumed by one CoTS during the 9 days (CoTS 5) and a minimum number of four *Acropora*—other colonies (CoTS 4 and CoTS 6). *Acropora*—other accounted for 51% of all coral colonies consumed during the study. The second most favoured coral taxon was *Montipora*, accounting for 18% of all coral colonies consumed, followed by *Pocillopora* (5%) and *Astreopora* (5%).

The relative contributions of different taxa to CoTS’ diet differ depending on how feeding rates were measured (Figure 4). When relative consumption of coral taxa is explored from a colony perspective (Figure 4a) *Acropora*—other accounts for 51% of all colonies consumed, *Montipora* accounts for 18% and *Pocillopora* 5%, however when relative consumption of coral taxa is explored from a 3D surface areal perspective (Figure 4b), *Acropora*—other accounts for 82% of all feeding and *Montipora* only accounts for 9%, followed by *Pocillopora* (3%) and *Acropora*—table (2%). The proportion of *Acropora*—other surface area consumed by each individual CoTS ranged from 41.6% of the diet (CoTS 4) to 99.4% (CoTS 3), with *Acropora*—other contributing to >80% of total coral area intake for 5 out of 8 CoTS. Acroporidae corals (which encompass *Acropora*, *Montipora* and *Astreopora* corals from this study) accounted for 94% of the coral intake in this study (3D surface area, all CoTS pooled).

### 3.4. Scar Ageing

Scars took between 3 and 7 days to surpass the chroma values associated with a Level 2 on the CoralWatch card (Figure 5b). Minimal colour change was observed in the initial days post-consumption, but from approximately days 3–5 onwards, algal growth increased markedly and continuously, eventually rendering colonies indistinguishable from surrounding reef features.

## 4. Discussion

Intensive research into the daily feeding rates of adult CoTS revealed marked variability in feeding activity across successive days. Most individuals displayed bouts of intense feeding interspersed with consecutive days of no apparent activity. In fact, feeding activity was only recorded on 65% of the nights within the observation period, with no individual starfish feeding on every night of the study period. This suggests that on any given day only ~65% of CoTS in a population might actually feed, which has consequences for their ecological impacts but also for detectability of CoTS based on occurrence of recent feeding scars [41]. Whilst consecutive days of inactivity were common, no CoTS remained inactive for longer than 3 consecutive days which may be the fasting limit for individuals in this size range. However, periods of inactivity did not necessarily correspond with the extent of feeding on preceding days, inferring that periods of inactivity did not appear to be directly motivated by satiation. For example, after a large feeding bout on day 4 of the study (471 cm^2^ planar area), CoTS 3 continued to feed at high rates for the following three consecutive nights, whereas CoTS 6 consumed a small quantity (50.2 cm^2^) and was inactive for the following 2 nights. This variability in behaviour suggests that rest periods and foraging behaviour are perhaps motivated by other factors than satiation. It may be that CoTS feed opportunistically based on local availability of different prey corals, until factors (i.e., predation risk, temperature or chemical cues) deter feeding.

This study revealed considerable variation in areal extent of coral consumed, over consecutive days and also among individuals. Daily feeding bouts were recorded as high as 9 colonies or 1261 cm^2^ day^−1^ (planar area) for a single starfish in a 24-h period. These values demonstrate the ability of individual CoTS to consume vast amounts of coral during 24-h periods, but such large feeding bouts were not sustained. Average daily feeding rates are much lower (198.39 cm^2^ day^−1^ ± 23.86 SE), mainly because they incorporate multiple days of no feeding, which is the most appropriate metric for establishing net coral loss. Our mean daily feeding rate calculated for all CoTS in this study closely corresponds with the feeding rate (231 cm^2^ day^−1^ ± 122 SE) reported by Foo et al. [42] for CoTS (*Acanthaster* cf. *solaris*) within the central and western Pacific.

Daily feeding rates are presented in three distinct formats: the number of colonies consumed, the projected planar (2D) area and the overall tissue (3D) surface area of coral consumed per day. These different metrics offer different perspectives on CoTS feeding; the number of colonies consumed provides insights into feeding activity and movement, the planar area estimates provide indication of the cumulative ecological impacts of CoTS feeding, whilst the overall 3D tissue surface area provides an improved understanding of potential differences in feeding efficiency between complex versus simple corals. This unique perspective on daily feeding provides insight into the physiological trade-offs of feeding choices. When cumulative feeding over the entire survey period was compared among CoTS, considerable differences between individuals were identified (Figure 4). It could be reasonably assumed that CoTS of the same species, subject to similar environmental conditions and living in comparable habitats, would exhibit very similar feeding activity and rates. However, pronounced differences among individual CoTS were found, especially when comparing the overall 3D surface area of coral consumed (Figure 4b), which may affect energetic intake.

Investigations into the drivers of inter-individual variation found no effect of body size of GSI on daily feeding rate of individuals, with individual identity accounting for the greatest proportion of observed variability. This contrasts with other studies that have found body size to influence feeding rates (cf. [22,43,44]), though focal individuals in this study showed only moderate variation in size (35–47 cm). Post hoc dissection confirmed that 7 of the 8 individuals were female with similar investment into gonad production (GSI), making it difficult to evaluate the influence of sex or reproductive status. The differences observed are therefore unlikely to be driven by the measured intrinsic factors and may instead reflect heterogeneity in temporal patterns of feeding, such that longer term studies would be needed to ascertain potential differences in overall energetic intake, which would affect individual condition and fitness. Due to broad similarities in the environmental conditions of the three survey sites, no significant differences between sites were recorded; however, it is possible that fine-scale habitat differences (e.g., local coral assemblage and habitat structure) may have influenced foraging behaviour to some extent. High-resolution spatial habitat data that was collected concurrently for the same individuals will help clarify the role of local habitat in shaping feeding activity, although these results are beyond the scope of the present study.

While potential drivers of feeding variation were explored within this study, identifying the influence of intrinsic (e.g., body size, sex) or extrinsic factors (e.g., coral availability, population density, season) was not the primary aim. Given the sample size and relatively homogeneous conditions, it is difficult to determine the effect of these factors on feeding rates from this study alone; however, ongoing work using larger sample sizes across different environmental and population contexts will help place these findings in a broader ecological context (Chandler et al., *in preparation*). Whilst a degree of caution should be exercised when extrapolating these findings to broader CoTS populations in contrasting environments, these results offer valuable insight into individual-level feeding behaviour under ecologically relevant conditions for the GBR, with direct application to local outbreak modelling and management. This study provides one of the most detailed empirical assessments of daily feeding by *Acanthaster* cf. *solaris*, and these insights can inform predictive tools for outbreak dynamics, improve estimations of coral loss, and refine thresholds for management intervention. Such information is essential for reef managers seeking to forecast and mitigate the impacts of CoTS on vulnerable reef systems.

In this study, diet composition was found to play a key role in shaping both daily and cumulative feeding rates. This is likely due to taxon-specific differences in morphological complexity that inherently influence the areal extent of coral tissue consumed. By accounting for differences in the overall (3D) tissue surface area consumed, this study revealed important taxonomic differences in the rates of 2D versus 3D feeding. In this study *Acropora*—other corals were consumed in excess of all other taxa, contributing disproportionately to total tissue area consumed by all CoTS combined (82% of all tissue area). This dominance could be a result of the local environment, where *Acropora* corals were abundant, but is also likely a result of the highly complex growth form of this genus, which offers a large surface area per colony consumed. Indeed, surface area coverage of *Acropora* tissue can be over five times that of encrusting corals of the same colony diameter [37], which may explain why this genus accounted for the majority of tissue area consumed, despite numerous colonies of other morphologically simple taxa being consumed (e.g., *Montipora* and *Astreopora*). Given their relatively high tissue (3D) surface area, highly complex corals such as *Acropora* presumably provide higher energetic return. However, tissue thickness, tissue accessibility, energetic content and digestibility will also influence overall energetic return [15]. Keesing [15] showed that *Acropora* corals have high energetic content and protein content, compared to other corals, which will further compound benefits provided by increased tissue surface area. These insights may explain the consistently high feeding preferences for this genus which have been well-documented in the literature (e.g., [18,45]). Given that Acroporidae corals as a whole accounted for 94% of total surface area consumed by all CoTS combined, this raises concerns about how CoTS will fare in future scenarios where these corals are forecast to diminish [29], and highlights the potential for significant shifts in coral community structure following outbreaks [31].

Daily tracking of CoTS feeding activity during this study provided a unique opportunity to document the progression of scar ageing over a 14-day period. Fresh feeding scars were observed to reach the colouration characteristic of ‘aged’ scars (defined as being overgrown with turf algae and comparable to colour chart B2) within 5–6 days during summer (Figure 5). These findings complement those of Cumming [46], who reported that coral injuries at this latitude typically begin to discolour within 3–5 days during summer. They also align closely with the observations of Biggs and Eminson [47], who found that CoTS scars became indistinguishable from surrounding coral features within two weeks, with rapid turf algal colonisation occurring within the first week post-predation. Understanding the timing of scar ageing enables more accurate estimation of daily feeding rates from large-scale field surveys.

## 5. Conclusions

This study resolved that the feeding activity of individual CoTS varies greatly among successive days. Most notably, it revealed that CoTS do not consistently feed in consecutive 24-h periods, but may remain inactive for periods of 2–3 days. This has implications for their ecological impact, but also affects detectability, as CoTS are often hidden during periods between feeding bouts. Highly resolved mean daily feeding rates presented herein were found to fit within the range of previous feeding studies, albeit at the smaller end of the continuum which is likely because multiple days of no feeding were captured within the 2-week monitoring period. These rates of feeding, which explicitly account for periods of inactivity, will be useful for updating ecological models of CoTS impact on reef assemblages. Further investigation into different intrinsic and extrinsic factors that drive variation in CoTS feeding will be useful for effectively predicting the ecological effects of CoTS in different regions and environmental settings. It is likely that starfish feeding behaviour is determined by a complex interaction of factors (including body size, prey availability and composition), and understanding these drivers is important for understanding and predicting their ecological effects in coral reef ecosystems.

## Figures and Tables

**Figure 1 biology-14-01001-f001:**
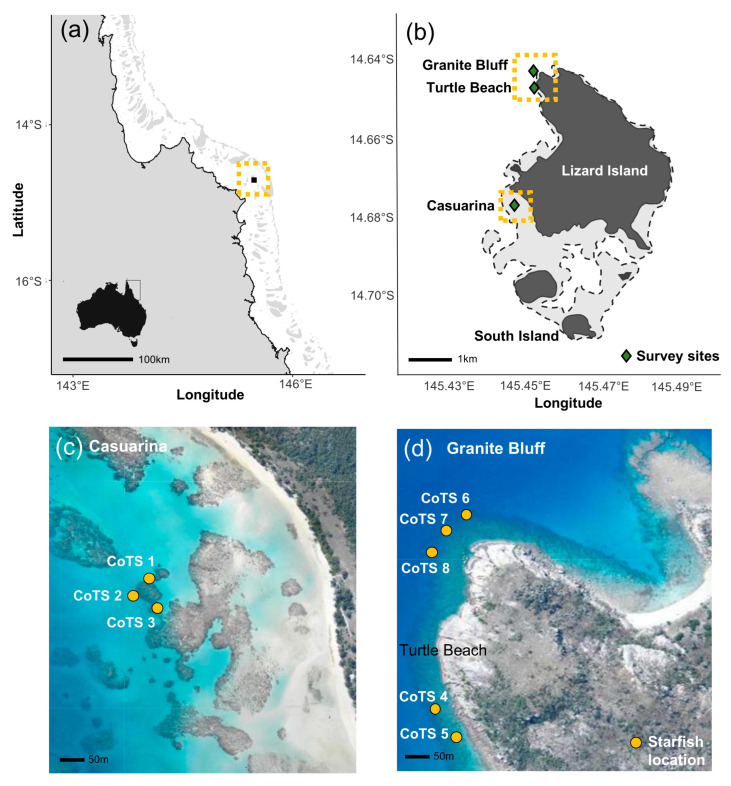
Maps of study location and sites. (**a**) Lizard Island’s location in the northern Great Barrier Reef, Australia, depicted by the yellow box; (**b**) Lizard Island with survey locations Granite Bluff, Turtle Beach and Casuarina marked in green, highlighted by yellow boxes; (**c**) the locations of CoTS 1–3 tracked at Casuarina; and (**d**) the locations of CoTS 4–8 at Turtle Beach and Granite Bluff. Queensland Imagery online aerial photograph library. Accessed 18 May 2024. https://qimagery.information.qld.gov.au/.

**Figure 2 biology-14-01001-f002:**
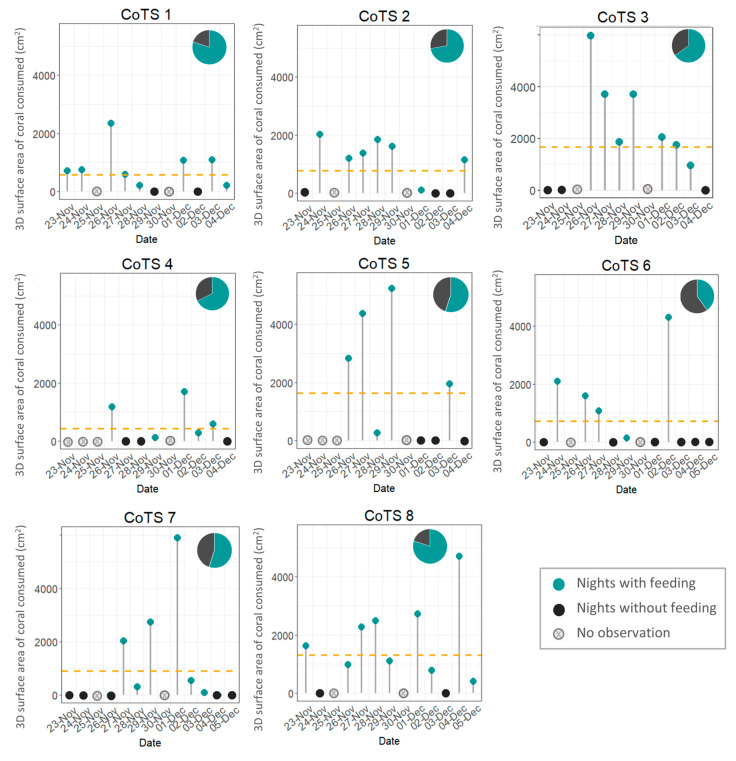
Daily activity patterns of 8 individual starfish over the survey period (9–13 days). Green points represent the 3D surface areal extent (cm^2^) of coral consumed during each 24-h period, grey points represent 24-h periods when no coral was consumed. Days when no feeding observations were conducted (due to diving limitations) are marked as crossed circles. Dashed lines represent the average number of colonies consumed over the survey period. Pie charts represent the proportion of 24-h periods spent feeding (green).

**Figure 3 biology-14-01001-f003:**
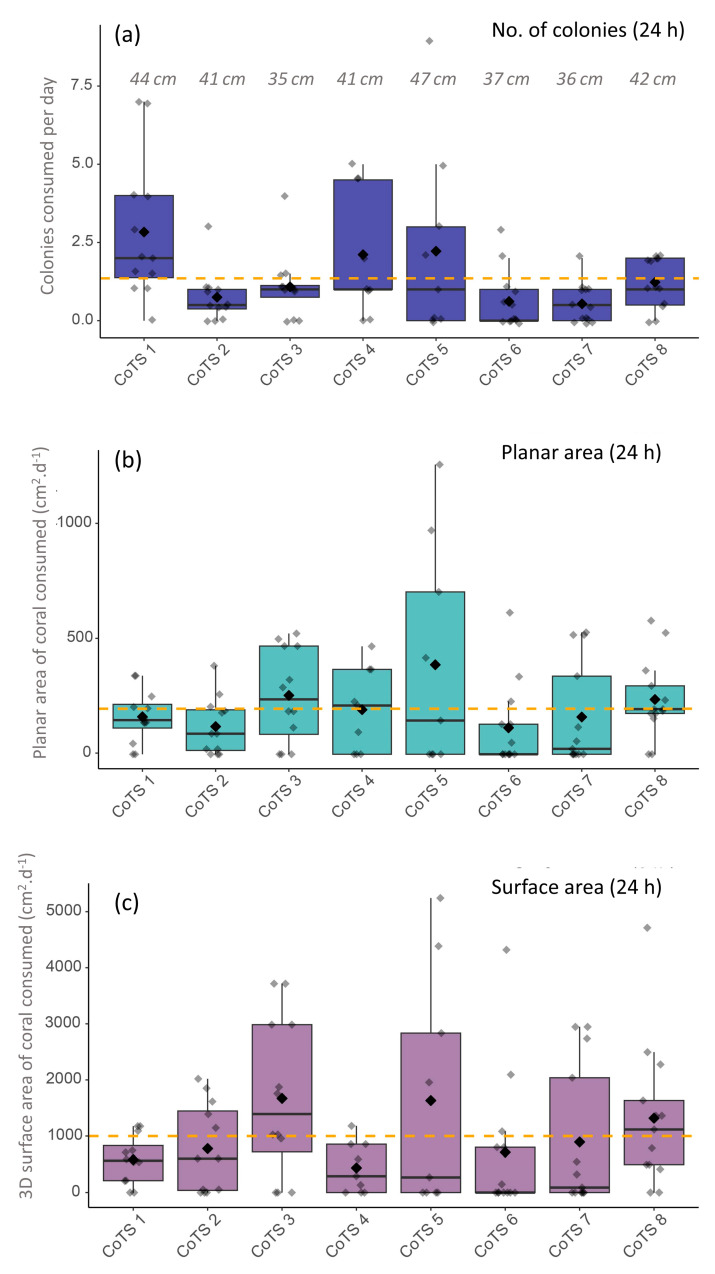
Daily feeding rates of 8 individual starfish recorded over 9–13 consecutive days, presented as (**a**) number of colonies consumed per day, (**b**) planar projected area of coral consumed per day (cm^2^) and (**c**) total (3D) surface area of coral consumed per day (cm^2^). Each box plot represents feeding for a single starfish across multiple days of intensive sampling. Large black diamonds and thick horizontal lines represent mean and median values, respectively. Maximum body diameter of each individual CoTS is also presented in panel **a**.

**Figure 4 biology-14-01001-f004:**
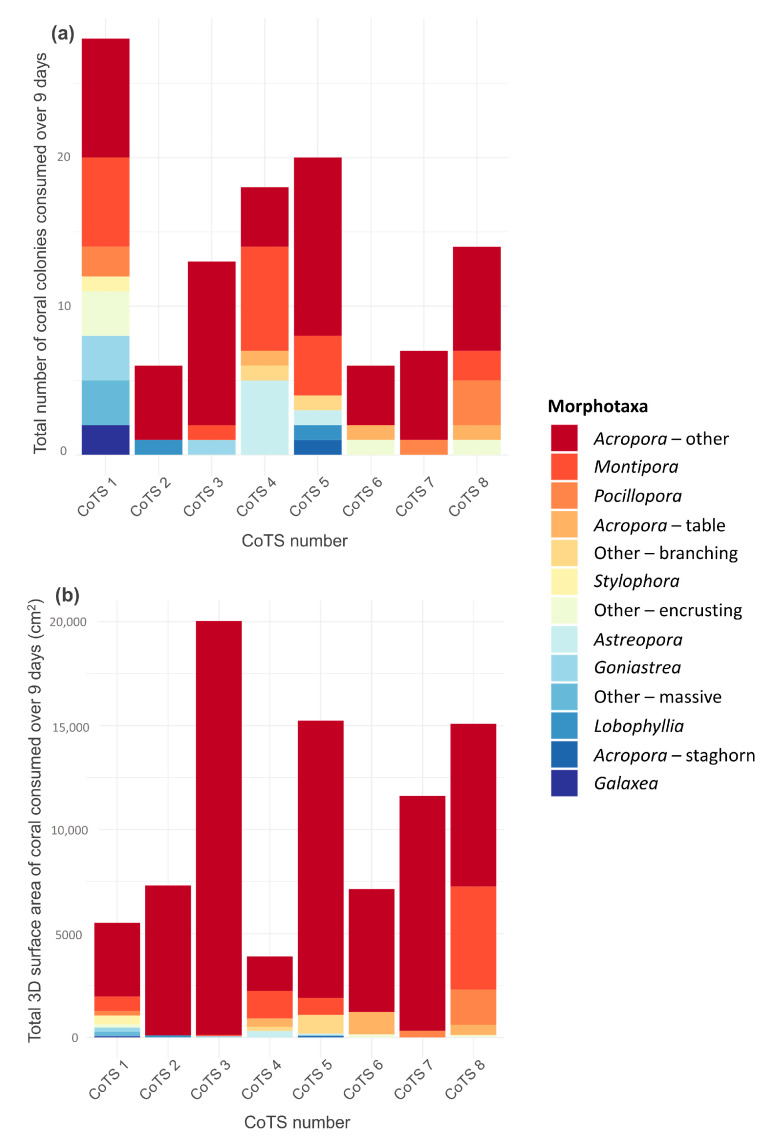
Total (cumulative) feeding from 9 consecutive days of feeding by 8 individual starfish. Bars represent summed feeding and colours depict proportional feeding on different coral morphotaxa. Total feeding is presented as (**a**) total number of coral colonies consumed and (**b**) total 3D surface area of coral consumed (cm^2^).

**Figure 5 biology-14-01001-f005:**
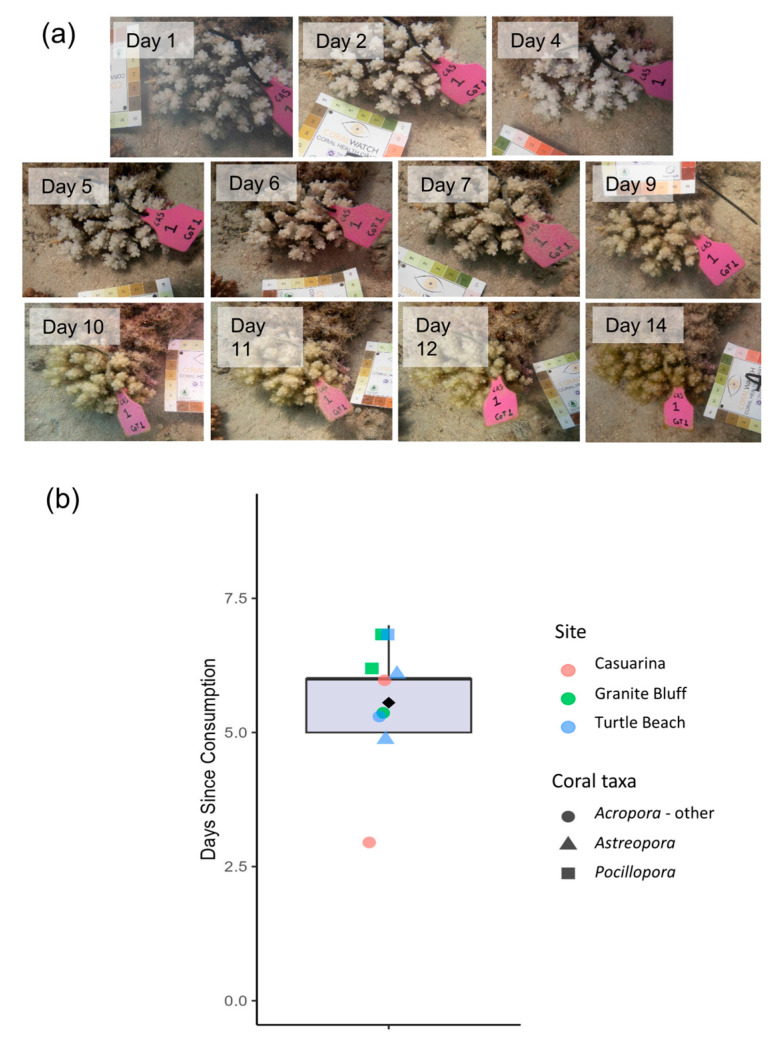
Ageing process of feeding scars observed. (**a**) Visual progression of a feeding scar (genus *Pocillopora*) photographed over 14 days in this study. (**b**) Box plot showing the number of days (since consumption) at which the B2 colour threshold was surpassed for 9 tracked scars. The box plot line shows the median time, the box limits show the interquartile range, the whiskers show the 1.5× interquartile range and the filled diamond shows the mean. Each coloured point represents an individual scar for which the ageing period was quantified. The shapes and colours of each point indicate the coral taxon and sampling site of the scars.

## Data Availability

All data analysed as part of this study are available from Research JCU (https://doi.org/10.25903/pqx9-sy53). The data will remain under embargo until the manuscript is published.

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
