# Peer review of "Daily Variation in the Feeding Activity of Pacific Crown-of-Thorns Starfish (Acanthaster cf. solaris)"

_biology, 2025, doi:10.3390/biology14081001_

Round 1
Reviewer 1 Report
Comments and Suggestions for Authors
This manuscript (MS) investigates the crown-of-thorns starfish (CoTS) to measure their feeding rate and patterns. CoTS feed irregularly but show consistent average daily consumption. Acropora corals form the majority of their diet, indicating that CoTS are highly dependent on this genus, which could affect their feeding success if Acropora populations decline. Overall, the MS is well-written and analyzed. However, I do have several minor comments that I believe need addressing before publication.
Abstract and introduction
- Line 29 or 51: Please correctly write the species name, Acanthaster spp . Please do for all species names.
Method
- Line 135. Please explain the specific reason why the authors chose Lizard Island as a research location.
- Line 136: Please explain regarding the starfish that was used for this experiment. It's originally from the research location or introduced from another region. Any tagging method was used for each starfish?.
- Did you consider environmental factors, e.g., temperature, etc, changes that are possible affect feeding behaviour?
Result
- The figure should display in better colour. Grey graphics are not easy to catch by the reader. I suggest using more colour-neutral and color gradients that are easily discernible by the eye.
- Line 264: In Figure 2, CoTS 4. There are look like overlapping markings in 26 and 27 November. Please re-check.
- Line 264: in Figure 2, CoTS 5. There is a high night without feeding (grey full cycles) value on 03 December. Please re-check the figure.
- Line 332. Can you increase photo sizes to make all the information that you serve easy to catch.
Discussion
- Can you elaborate on the possibility that the position of the exp. station (near or far from the shore, coral abundance, or competition are also affect starfish feeding behavior or rate?
Reviewer 2 Report
Comments and Suggestions for Authors
Dear Editor and Authors,
The reviewed manuscript presents the results of an in situ underwater study on the feeding behavior of the crown-of-thorns starfish (COTS) in Australia. The manuscript is generally well written and the English is of high quality. Conducting underwater studies of this nature are challenging. However, there are several methodological shortcomings and omissions that must be addressed —if possible— before the manuscript can be considered for publication.
Major Concerns:
- The number of observed individuals (n = 8) is quite low.
- Observations were conducted during only a single season, which is not clearly defined.
- No environmental data (e.g., temperature, salinity) are presented or discussed.
- Spatial variation across the different study sites is not considered and included in the analysis.
- Variation in feeding behavior across different coral species is not analyzed.
These issues must be addressed, relevant data included (if available), and methodological choices explicitly justified in order to support the validity of the study’s conclusions. Furthermore, the ecological significance of the results is not well presented, making it difficult to evaluate the broader context and impact of the findings.
For these reasons, I recommend “major revision”. I hope the authors possess the necessary data and expertise to address these concerns.
Specific Comments and Suggestions:
Introduction
This section is generally well written with adequate background and references. Please include information about the geographic distribution of Acanthaster spp. Is the species native to northern Australia? Is it invasive or widespread globally? This would help readers unfamiliar with the region understand the ecological importance of the species.
Line 51: Please provide the full taxonomic name and use italics the genus throughout the manuscript.
Lines 115–118: This sentence is overly long and a bit unclear. Consider breaking it into two for clarity.
Materials and Methods
As noted above, several key methodological details are either missing or insufficiently justified:The sample size (n = 8) is likely too small to allow robust generalization. While underwater research is inherently logistically constrained, the authors should justify why this number is considered adequate.
- The study was conducted in only one season, which is not identified. Results may be season-specific, and this limitation should be acknowledged and discussed.
- Environmental factors such as daily water temperature fluctuations, light levels, or current velocity—known to influence feeding behavior—are not mentioned or analyzed.
- Variation among study sites (in environmental characteristics and coral cover) is not evaluated.
- Please include underwater photographs and detailed descriptions of the study locations. If possible, also include images of the COTS feeding.
Line 136: Please clarify whether 8 individuals are enough to yield meaningful conclusions.
Line 141: The term “high coral cover” is vague—please quantify and compare coral cover across sites. Site size should also be reported.
Lines 148–149: The description of tracking duration is confusing. Please rephrase or clarify.
Lines 148–156: Consider using a table or figure to summarize the movement/tracking information more clearly.
Line 241: Please list the factors considered in the ZIGLMM and the assumptions of the model.
Results
The figures are clear and easy to follow. However, the ZIGLMM analysis which appeared to be a central part of the study, is not well properly explained or interpreted. If this analysis does not contribute meaningful insights, it should be removed. If it is important, its results should be more clearly presented and discussed.
Discussion
- The discussion is generally well written and addresses most of the study’s findings (except for the ZIGLMM, which should be discussed in more detail or omitted).
- The ecological relevance of the results is only briefly mentioned in the conclusion. This aspect should be expanded within the discussion to emphasize the significance of the findings in a broader ecological or conservation context.
- Once methodological concerns are addressed, the discussion should be updated accordingly to reflect the new or revised results.
Round 2
Reviewer 2 Report
Comments and Suggestions for Authors
All points raised have been addressed